



# Large-eddy simulation of traffic-related air pollution at a very high-resolution in a mega-city: Evaluation against mobile sensors and insights for influencing factors

Yanxu Zhang[1], Xingpei Ye[1], Shibao Wang[1], Xiaojing He[2], Lingyao Dong[1], Ning Zhang[1], Haikun Wang[1], Zhongrui Wang[1], Yun Ma[1], Lei Wang[1], Xuguang Chi[1], Aijun Ding[1], Mingzhi Yao[3], Yunpeng Li[3], Qilin Li[3], Ling Zhang4, Yongle Xiao4

[1]School of Atmospheric Sciences, Nanjing University, Nanjing, China
[2]School of Environment, Nanjing University, Nanjing, China
[3]Beijing SPC Environment Protection Tech Company Ltd., Beijing, China
4Hebei Saihero Environmental Protection Hi-tech. Company Ltd., Shijiazhuang, Hebei, China
*Correspondence to*: Yanxu Zhang (zhangyx@nju.edu.cn), Ning Zhang (ningzhang@nju.edu.cn), and Haikun Wang (wanghk@nju.edu.cn)

**Abstract.** Urban air pollution has tremendous spatial variability at scales ranged from kilometer to meters due to unevenly distributed emission sources, complex flow patterns, and photochemical reactions. However, high-resolution air quality information is not available through traditional approaches such as ground-based measurements and regional air quality models (with typical resolution >1 km). Here we
develop a ten-meter resolution air quality model for traffic-related CO pollution based on the parallelized large-eddy simulation model (PALM). The model performance is evaluated with measurements obtained from sensors deployed on a taxi platform, which collects data with a comparable spatial resolution to our model. The very high resolution of the model reveals a detailed geographical dispersion pattern of air pollution in and out of the road network. The model results ($0.92 \pm 0.40$ mg/m$^3$) agree well with the
measurements ($0.90 \pm 0.58$ mg/m$^3$, $n = 114,502$). The model has similar spatial patterns with that of the measurements, and the $r^2$ value of a linear regression between model and measurement data is $0.50 \pm 0.07$ during non-rush hours with middle and low wind speeds. A non-linear relationship is found between average modeled concentrations and wind speed with higher concentrations under calm wind speeds. The modeled concentrations are also 20-30% higher in streets that align with the wind direction within ~20°.
We find that streets with higher buildings in the downwind have lower modeled concentrations at the pedestrian level, and similar effects are found for the variability in building heights (including gaps between buildings). The modeled concentrations also decay fast in the first ~50 m from the nearest highway and arterial road but change slower further away. This study demonstrates the potential of large eddy simulation in urban air quality modeling, which is a vigorous part of the smart city system and could inform
urban planning and air quality management.

## 1 Introduction

Urban air pollution is one of the greatest threat for human health in the modern world as 55% of the global population are living in cities but more than 80% of them are exposing to air quality levels that exceed the World Health Organization limits(The World Bank, 2020; WHO, 2016). Traffic related emissions are often
the major source for urban regions for many air pollutants (e.g. CO, nitrogen oxides, and volatile organic compounds)(Liu and He, 2012). Patterns of traffic-related air pollution in the urban environment has substantial temporal and spatial variability due to unevenly distributed emission sources, complex flow pattern, and physicochemical transformations(Apte et al., 2017). Compounded with the complex and dynamic commuting behavior and crowd dynamics of urban residents, high-resolution air quality
information is thus needed for smart-city designers and air pollution mitigation in a "big-data" era(Gao et al., 2019). However, such information is generally not available as accurate ground-based monitoring of air quality at a high spatial resolution is too expensive due to the large number of required instruments even with relatively low-cost sensors(Kumar et al., 2015). The typical monitoring site numbers are ~10 even for a megacity with >10 million population and >1000 km$^2$ areas, and these sites are often located far away
from road networks. Alternative approaches such as satellite remote sensing and regional chemical transport models are also spatially coarse (~1-10 km resolution)(van Donkelaar et al., 2010; Zhang et al.,





2009). Here we present a very high-resolution air quality model for traffic-related air pollution in urban regions using large-eddy simulation.

The impact of traffic emission on urban air quality is associated with a myriad of factors such as emission strength and air pollutant dispersion(Abou-Senna et al., 2013). For example, background meteorological factors such as the wind speed and vertical temperature stratification are known to influence the pollutant dispersion, and the most severe air pollution is associated with calm weather conditions with temperature inversions(Wolf and Esau, 2014). Trees are found to increase turbulence and reduce ambient concentrations associated with traffic emissions at pedestrian height(Jeanjean et al., 2015). The geometry of the street
canyon is an important factor: higher buildings and narrower streets cause heavier pollution inside canyon(Fu et al., 2017). The symmetric level of building heights also influence wind and turbulent diffusion and affect pedestrian level concentrations(Fu et al., 2017). Preferable pathways created by the configuration of buildings and streets facilitate longer dispersion of pollutants and influence regions farther away from roads(Wolf et al., 2020).

Numerical models have been applied to model traffic-related air pollution in urban regions. Gaussian models have been widely used in such purpose for a long history, e.g. regulatory models such as AERMOD and CALPUFF(US EPA, 2020). These models use statistical method to parameterize turbulent diffusion based on background meteorological conditions and diagnostic building geometry characteristics, and reasonable accurate results can be achieved with representative meteorological input(Rood, 2014).
Dispersion models are also nested with regional Eulerian models such as CMAQ and CAMx to bridge the coarse resolution (~km) to street-level (~10 m) (e.g. the ADMS-Urban model(Biggart et al., 2019; Righi et al., 2009)). One drawback of these statistical models is the lack of explicit representation of the air flow and turbulent eddies around landscape and buildings(Sun et al., 2016). The predicting power of these models decreases farther away from sources as they cannot describe the turbulent transport of pollutants by larger
eddies which could trap air parcels over longer distances(Wolf et al., 2020). In recent years, computational fluid models (CFD) that are turbulence-resolving or permitting have been used for urban air quality purpose, starting from ideal conditions(Kurppa et al., 2018; Sanchez et al., 2005; Steffens et al., 2014; Yu and Thé, 2017) to city-wide simulations(Cécé et al., 2016; Jeanjean et al., 2015; Wolf et al., 2020). For instance, Sanchez et al.(Sanchez et al., 2005) simulates reactive pollutants (NOx, VOC, and $O_3$) and their
reactions in an urban street canyon using the OpenFOAM model. Wolf et al.(Wolf et al., 2020) utilizes the Parallelized Large-Eddy Simulation Model (PALM) to simulate $NO_2$ and $PM_{2.5}$ air quality in a coastal city, and successfully identified major sources under high pollution meteorological conditions.

While the high-resolution models map urban air quality at street level, the tremendous high flow of spatial-resolved data are generally lacking proper evaluation against observations. Time series of pollutant
concentration data from a limited number of stationary stations are often used to compare to the model results(Biggart et al., 2019; Cécé et al., 2016; Fu et al., 2017). For instance, Biggart et al.(Biggart et al., 2019) compared their model predictions at street-scale-resolution to eight stations across the city of Beijing with a model domain area of ~400 $km^2$. Even though a good correlation is often achieved in these studies, the success in predicting temporal variability does not automatically transfer to spatial variability. In this
study, we develop a very high spatial resolution (less than 10 m) model for traffic-related air quality based on the PALM model for the city of Nanjing, a megacity in eastern China with more than eight million population. We evaluate the model performance with observations obtained from sensors deployed on taxi platforms, which garner data with comparable spatial resolution to the PALM model. Multiple influencing factors for pedestrian-level air pollution levels are also investigated.

## 2 Methodology

### 2.1 PALM Model

We use the PALM model system to simulate the transport of traffic-related emissions in Nanjing. This model is developed by the PALM group at the Leibniz University of Hannover, and has been developed as a turbulence-resolving large-eddy simulation (LES) model system especially for performing on massively
parallel computer architectures. We use PALM 4 (version number 3689) for urban applications in this study(The PALM Group, 2020), which includes a dynamic solver for the Navier-Stokes equations and the first law of thermodynamics. The bulk of the turbulent motions in the atmospheric boundary are explicitly resolved(The PALM Group, 2020). To save the model computation time, the pollutants are considered as a passive scalar (i.e. no chemical reactions and deposition), and a neutral stratification condition is assumed



(i.e. no buoyancy related terms are calculated). The 5[th] order upwind scheme of Wicker and Skamarock is used for both momentum and tracer advection(Wicker and Skamarock, 2002). We use CO as a representative pollutant as its relatively long lifetime (months to years)(Jaffe, 1968). So the chemical reactions and dry and wet deposition are generally negligible within the time scale of model simulation (hours). Neumann type boundary condition is used for the chemical tracer, and a Dirichlet type one is for

velocities.

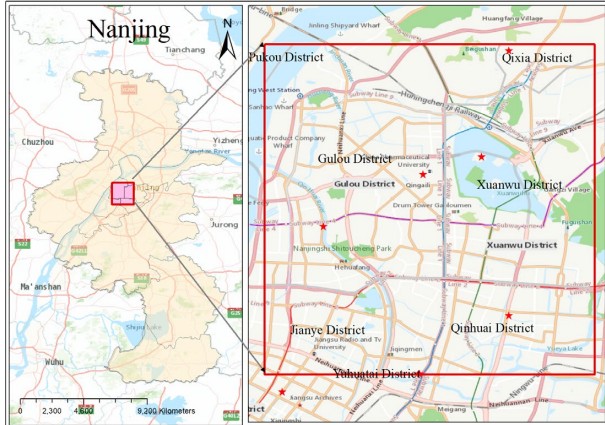

Figure 1. Model domain of the PALM model simulation used in the study for the city of Nanjing. The size of the model domain is approximately 10 km × 10 km. Map credit: ESRI 2020.

The model domain covers the core area of Nanjing with the center located at 32.07°N and 118.72°E (Figure

1). The model horizontal resolution is 0.0001°×0.0001° (equivalent to 9.4 m west-east ×11.1 m north-south) with a grid size of 960×960, which covers a total area of about 10 km × 10 km. To represent the air quality at pedestrian-level, the model vertical layer depth starts with 2 m from the ground to 12 m height, and stretched by a factor of 1.1 by each layer to a maximum of 40 m depth. The model has a total of 48 vertical layers reaching ~1000 m a.s.l., which is approximately three times higher than the highest building of

Nanjing (Zifeng Tower, 340 m height for the top floor). The model is run for three hours with a time step of 6 seconds. Hourly average data is achieved and we use the results of the last hour for analysis.

The topography of the model consists of two parts: baseline elevation and building heights. The former is based on ASTER global digital elevation model (GDEM) dataset, which has a native resolution of 30 m and is linearly interpolated to the model grid(Computer Network Information Center, 2020). The building

data for Nanjing is extracted from Gaode Map (dated as the year 2018, https://ditu.amap.com). The building data includes the geographical location of the outer shape of buildings and their number of floors. We transfer the raw data into the model grid and assume an average floor height of 3 m (Figure 2A). The sum of the elevation and building height data are then used as the topographical data of the PALM model. Due to the large computational cost associated with model simulation, we choose to run the model only for

a selective combination of meteorological scenarios. For each scenario, we assume a constant geostrophic wind field on the top of boundary layer during model simulation. Eight wind directions with 45° apart (N, NE, E, SE, S, SW, W, and NW) are considered. Based on the observed wind speed at the top of local boundary layer (~500 m)(Chen et al., 2018; He et al., 2018), we choose 10 m/s, 6.5 m/s, and 3 m/s to represent high, median, and weak wind conditions, respectively.

**2.2 Traffic Emissions**

We use a "standard road length" apprach to assign the total traffic emissions to individual road based on different road types and traffic flows(Zheng et al., 2009). We first transfer the actual road length (*L*) into total standard road length (*TSL*, km) of Nanjing using road conversion coefficient (*W*):

$$TSL = \sum_{j=1}^{o} \sum_{i=1}^{m} \sum_{k=1}^{n} L_{i,j,k} \times W_{i,j,k}$$      (1)



where $i$, $j$, and $k$ represent the area types (i.e., urban and suburban areas), grid cell index, and road type, respectively, with $m$, $n$, and $o$ representing the total numbers of area types, grid cells, and road types, respectively, and the $W$ is calculated as:

$$W_{i,j,k} = \frac{TF_{i,j,k}}{STF} \qquad (2)$$

where $TF_{i,j,k}$ is the traffic flow for the $k$th road type and $i$th area type in grid $j$ (in standard vehicles) and $STF$
is the standard traffic flow (in standard vehicles).

The traffic emission ($GE_j$) of each grid cell $j$ is calculated based on total standard road length in the grid cell ($GSL_j$) and the standard emission intensity per standard unit length ($SEI$, t/km):

$$GE_j = GSL_j \times SEI \qquad (3)$$

where $SEI$ is calculated based on the $TSL$ calculated in equation (1) and the city-level-based vehicle
emission inventories ($E$, t):

$$SEI = \frac{E}{TSL} \qquad (4)$$

and $GSL_j$ is calculated as:

$$GSL_j = \sum_{i=1}^{m} \sum_{k=1}^{n} L_{i,j,k} \times W_{i,j,k} \qquad (5)$$

We also assign the daily mean $GE_j$ to each hour based on the diurnal variation of the 24-hour traffic flow
(Figure 2B).

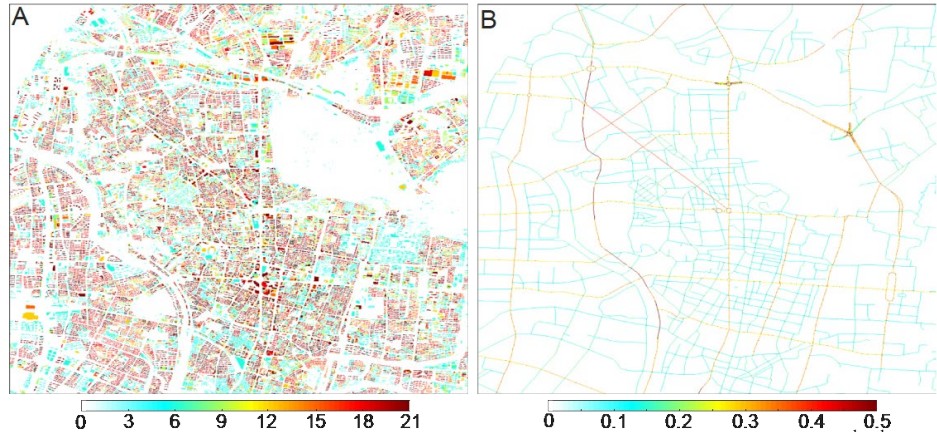

Figure 2. Spatial distribution of (A) building heights (m) and (B) traffic CO emissions (mg cell$^{-1}$ s$^{-1}$) (during rush hours) in the model domain.

**2.3 Taxi Sensor Data**

We evaluate the model results with observations collected from a mobile platform. The details of the platform instrument and its deployment are described with detail in the companion paper of this study(Wang et al., 2020). Briefly, we use two XHAQSN-508 instrument (dimensions: 290×81×55 mm; weight: 1.0 kg) produced by Hebei Sailhero Environmental Protection High-tech Co., Ltd. (Hebei, China), which includes an internal CO gas sensor (detectable CO range: 0 to 50 mg m$^{-3}$) and is installed on the top
of two Nanjing taxis (~1.5 m above ground). The sensor is capable of continuous measuring CO concentrations at a programmable frequency of once per 10 s. The inlet system is also optimized to minimize self-sampling and gas sampling losses. The spatial coordinates are also recorded by a GPS device included in this instrument (U-blox, Switzerland). The monitoring and location data are simultaneously transmitted to a remote server in real time through wireless communication, and the real time measurement
data can be viewed through a web page or an Android app. One major advantage of this mobile platform is the minimum maintenance cost, as samples are automatically collected during the operation of the taxis. An analyze of the sensing power, defined as the fraction of city road network sampled by a taxi fleet, also



demonstrates that a remarkably small number of taxis can scan a large number of streets(O'Keeffe et al., 2019; Wang et al., 2020).

The instrument is calibrated once per month against an stationary instrument (T300 CO Analyzer by Teledyne API) at the SORPES observation station in the Xianlin Campus of Nanjing University (https://as.nju.edu.cn/as_en/obsplatform/list.htm). During calibration, the instrument is taken back to the campus and placed back-to-back to the calibrating instrument in the station. The calibration lasts for at least seven days, and the parameters for the sensor retrieving algorithm are adjusted to make sure the differences

between the sensor retrieved data and the station data is < 1% (Wang et al., 2020). As only traffic-related emissions are considered in the PALM model, we add the model results to the background concentrations of Nanjing for comparing to the observed data by the mobile platform (but the pure model output is used for other analysis). The hourly background CO concentrations are calculated as the minimum of measurements from all the nine national air quality monitoring stations in Nanjing metropolitan area

(http://beijingair.sinaapp.com/). Corresponding hourly meteorological data of Nanjing city is obtained from the National Meteorological Information Center of China (http://data.cma.cn/user/toLogin.html).

**3 Results and Discussion**

**3.1 Very High-Resolution Air Quality Map**

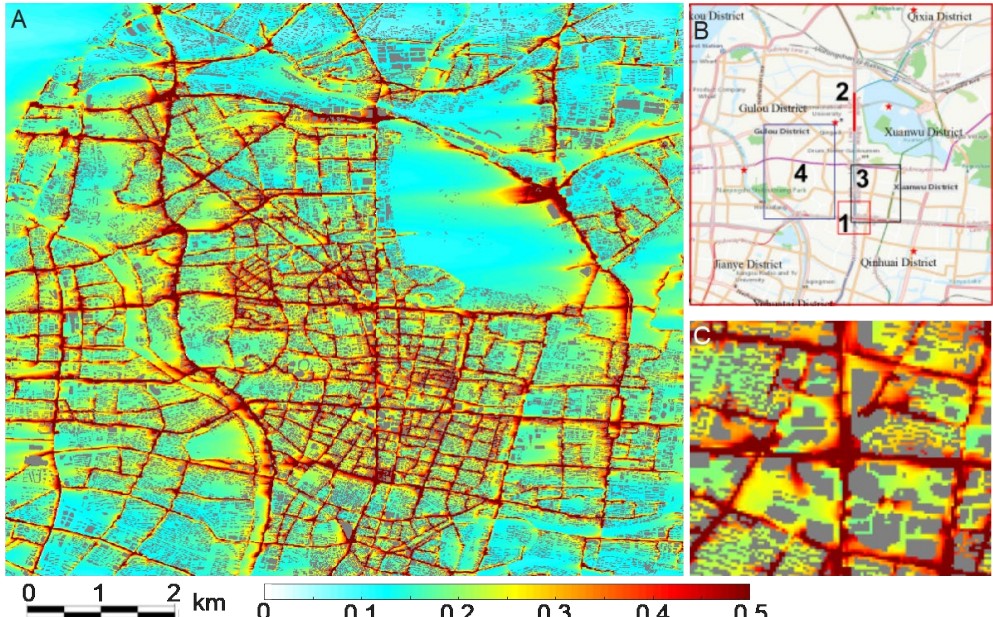

Figure 3. Modeled CO concentrations (mg/m$^3$) during rush hours by the PALM model with wind from the east and speed as 6.5 m/s in the top of the boundary layer (A). Panel B shows the corresponding city map. Panel C shows a zoom in over Xinjiekou area with boundary shown as a red rectangle in panel B (Rectangle 1). Grey areas represent top of buildings. Map credit: ESRI 2020.

Figure 3 shows an example of the spatial distribution of the modeled traffic-related CO concentrations

during peak hours (with an east wind and 6.5 m/s at the top of boundary layer). The very high resolution of the model reveals a detailed geographical dispersion pattern of CO concentrations in and out of the road network. The average modeled CO concentrations inside the road network are 0.76 mg/m$^3$ (with 25% and 75% percentiles as 0.45 – 0.94 mg/m$^3$, respectively), which are much larger than those outside the network: 0.22 (0.14 – 0.24) mg/m$^3$. The lowest concentrations are modeled over regions with less dense road

network and water bodies (~0.1 mg/m$^3$). Higher concentrations are modeled over major highways with substantially higher emissions than other roads (Figure 2B). The concentrations are also higher over interceptions of roads as the emissions are specified as the sum of that of the intercepted roads. The model simulates clear plumes downwind of major roads, especially if no obstacles existed in that direction. The



most apparent plume is simulated in the northeast of the Xuanwu Lake (refer to the map in Figure 3B). The
high emissions are swept for about 1 km westward from a traffic center at the northeast edge of the lake.
Highways such as the Neihuanxi Line also produce apparent westward plumes, whereas downwind
buildings may cause extra turbulence to smoothen out the signal. By contrast, the emissions from regions
with dense buildings are generally trapped within the street canyons (e.g. the city center), with leakage
from gaps between buildings (Figure 3C). Overall, the modeled concentrations follow a two-mode
Gaussian distribution (Figure 4), with one for residential streets (with a geometric mean of 0.17 mg/m$^3$) and
the other for arterial roads, highways, and the nearby regions (with a geometric mean of 0.28 mg/m$^3$).

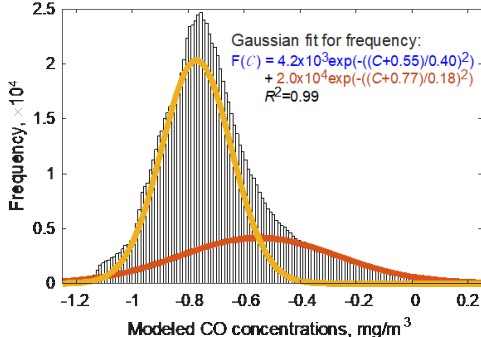

Figure 4. Frequency distribution of modeled CO concentrations during rush hours under east wind and 6.5
m/s at the top of boundary layer. The distribution is fitted with a two-mode Gaussian model. The
concentration is in standard logarithm scale.

**3.2 Model Evaluation**

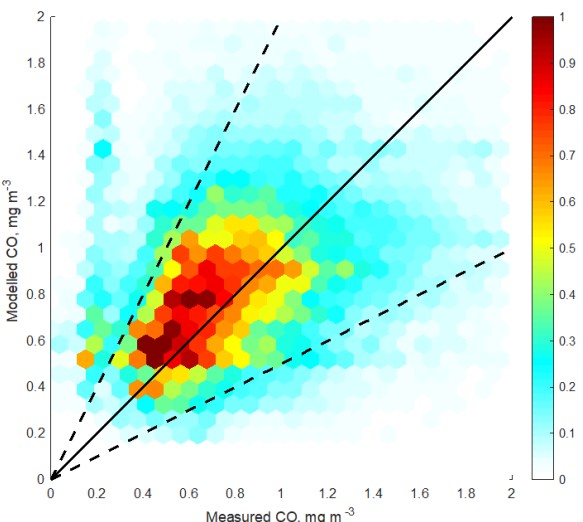

Figure 5. Comparison of measured and modeled CO concentrations. Colors represent the total number of
matching measured and modelled values contained within distinct hexagons. Black line indicates 1:1 and
dashed lines mark a factor of 2 difference.

The rich information provided by the model is compared to observations obtained by the mobile monitoring
platform (Figure 5 and 6). We sample the model results with the same location, time (rush or non-rush
hours), and meteorological conditions. The sum ($0.92 \pm 0.40$ mg/m$^3$) of model results that are caused by
traffic-related sources ($0.36 \pm 0.32$ mg/m$^3$) and regional background concentrations ($0.56 \pm 0.28$ mg/m$^3$)
agree well with the measured CO concentrations ($0.90 \pm 0.58$ mg/m$^3$, $n = 114,502$) ($p < 0.01$). This



indicates that traffic-related sources contribute ~40% of CO observed in the road network, while the contribution falls to 28% in other areas. Bottom-up emission inventory suggests that on-road transportation contributed ~11% of total CO emissions from Nanjing in 2012(Zhao et al., 2015). Considering the number of cars have increased by a factor of 2-3 since 2012 and the total CO emissions remained relatively

stable(Bureau Statistics of Nanjing Municipal, 2020), our results agree reasonably well with the inventory. Point-by-point comparison reveals that most of the data points fall near the 1:1 line and are within lines for a factor of 2 difference (Figure 5).

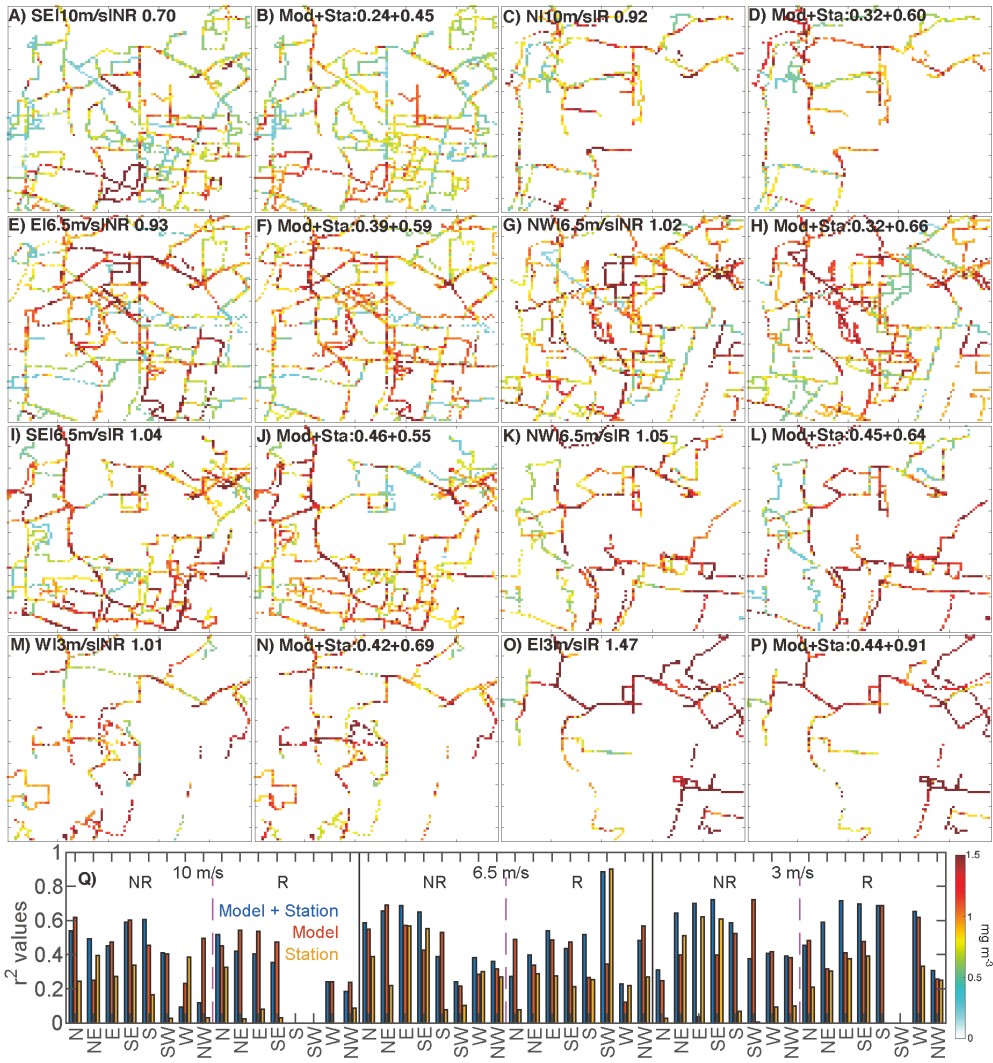

Figure 6. Comparison between taxi sensor measured (odd columns) and modeled (even columns) CO

concentrations for selected combinations of wind speed, directions, and rush/non-rush hours. As the taxi sensor data has a temporal resolution of 10 s (roughly equivalent to 100 m given an average vehicle speed of 40 km/h), both the measurements and model results are regraded to a 100 m resolution grid. The wind and emission information is shown on top of panels in this format: "wind direction | wind speed | emission level rush or non-rush hours)". The mean of the data is shown on top of each panel, with the modeled one

as the sum of the model output and regional background from national stations. Panel Q shows the coefficients of determination ($r^2$) of a linear regression between taxis sensor data and model/station data

under different emission and meteorological conditions. Blue bars represent the regression with model + regional background, while red and yellow bars are for the model and station data only, respectively.

As both the modeled and measured CO concentrations vary drastically, we group the data based on the sampling time and meteorological conditions and compare the spatial patterns of model results and the measurements in Figure 6 (more comparisons are available in the Supporting Information). We find the model captures many of the observed spatial features under a variety of emission and meteorological conditions. Take 10 m/s east wind during non-rush hours as an example (Figure 6A and 6B), higher concentrations are modeled and measured in the city center, the highway in north city, and the arterial roads

in the southwest corner of the model domain, while lower concentrations are in the middle of the west part and southeast corner of the model domain. Similar levels of agreement between the spatial patterns of measurements and model results are achieved for other conditions.

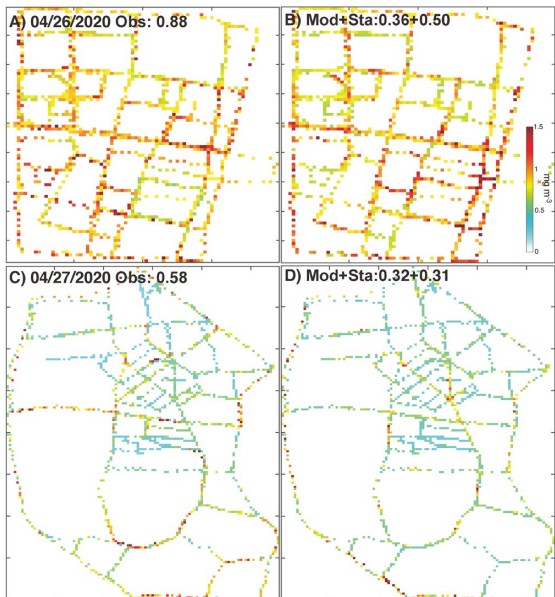

Figure 7. Comparison between taxi sensor measured (A and C) and modeled (B and D) CO concentrations
during two intensified observation campaigns during April 26-27, 2020. The locations of the campaigns are shown in Figure 3B (Rectangle 3 and 4 for the 26th and 27th, respectively).

Figure 6Q shows that the coefficients of determination ($r^2$) are generally higher (0.51 ± 0.16) during non-rush hours with middle and low wind speeds, due to the relatively larger sample sizes under such conditions (Supporting Information). The $r^2$ values for high wind condition and rush hours will be increased as the
accumulation of taxi sensor data (either longer sampling period or more sensors). As the model data used in this comparison includes the regional background, we calculate the $r^2$ values if only using the station data to rule out the possibility that the agreement in spatial pattern is caused by station data. Also taking the $r^2$ values during non-rush hours with middle and low wind conditions as an example, only using station data lower the $r^2$ values to 0.28 ± 0.23. This indicates that our model indeed carry useful spatial information that
significantly improve the comparison with sensor data.

One drawback of the taxi platform is that the popular streets are easily covered and sampled repeatedly, but unpopular segments are rarely visited(O'Keeffe et al., 2019; Wang et al., 2020). The sensor data used in this study mainly cover the highway and arterial roads, but generally leave the model results for residential streets unevaluated. We therefore supplement the routine taxi operation data with two one-day taxi cruise
campaigns, which cover all the public roads in two representative regions (especially including the residential ones less visited by taxies), as shown in Figure 7 (the location of campaign is shown in Figure 3B). Overall, the model captures the observed spatial patterns reasonably well with $r^2$ values for the two campaigns as 0.50 and 0.37, comparable to the data collected during normal taxi operations (Figure 6Q).





The first campaign is in the city center (Figure 7A and 7B) with the traffic-related CO concentrations
relatively more uniformly compared to the second one, which covers a larger area and includes highways,
        arterial, and residential roads (Figure 7C and 7D). The model also captures the relatively higher
        concentrations in the highway near the west edge in the second campaign (Figure 7C and 7D), as well as
        the generally decreasing concentrations from highways, arterial roads, to residential ones. Even though the
        model has highly simplified setting-ups and the mobile sensors have relatively large uncertainties compared
with reference method(Wang et al., 2020), the agreement between them lend both approaches confidence.

### 3.3 Influencing Factors

### 3.3.1 Emissions, wind speed and directions

Figure 8 shows the mean ground level CO concentrations over the whole model domain under different
emission strengths and meteorological factors. We find the wind speed is an important controlling factor for
modeled CO concentrations. The average CO concentrations during rush hours with a wind speed of 3 m/s
        range 0.37-0.46 mg/m$^3$. The concentrations with 3 m/s wind are ~2.4 and ~1.8 times higher than those with
        10 m/s ($0.16 - 0.19$ mg/m$^3$) and 6.5 m/s ($0.21 - 0.25$ mg/m$^3$) wind speeds, respectively. The concentration
        differences between 10 m/s and 6.5 m/s are about 30%. It clearly suggests a non-linear dependence of
        concentrations on wind speed with much higher concentrations over stagnant conditions, consistent with
previous studies(Mumovic et al., 2006; Wolf et al., 2020). Indeed, convective transport of pollutant is
        greatly reduced under low wind speed conditions, which elevates CO concentrations at the pedestrian level.
        On the other hand, the response to emission strength is almost linear with concentrations during rush hours
        are 27% higher than non-rush hours given the same meteorological conditions. The concentrations with
        different wind directions range ~20%, with consistent highest concentration for west wind and lowest for
northeast wind. This pattern could be explained by the spatial pattern of emission distributions: with higher
        emissions in the west part of the model domain and lower over the northeast (where a big lake locates).
        Wind from cleaner regions (e.g. northeast) helps to blow out the traffic-related emission located at the other
        side of the model domain, and vice versa.

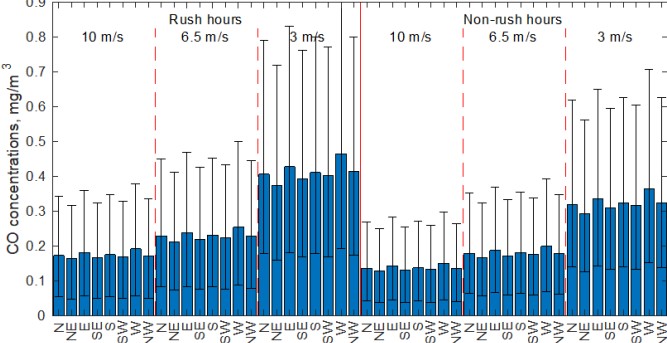

Figure 8. Mean modeled CO concentrations (with 30% and 70% percentiles) over the whole model domain
        with different wind speed, directions, and emission levels.

### 3.3.2 Street direction

Even though the wind direction seems not to be an important influencing factor for model domain-average
concentrations, it is a vital factor for individual street canyons. Figure 9 shows the relationship between the
mean CO concentrations and the angle between wind and street directions. We find the modeled
        concentrations are the highest when the wind direction aligns with streets. The concentrations decrease
        until the angle increases to ~20° but no significant differences are modeled when the angles continue to
        increase. A wind direction parallel with the street mainly transports CO along the canyon, which traps
        pollutant inside of the street. By contrast, a perpendicular wind can blow pollutant outside of the canyon
through gaps between buildings, which reduces the CO concentrations inside. Similar results have been
        found in smaller scale studies. For example, through comparing pollutant levels with different wind
        directions, Kurppa et al. (Kurppa et al., 2019) found lower pedestrian-level pollution when wind direction
        is closer to perpendicular with a boulevard and suggested the shortest wall parallel to the road to increase
        ventilation and create optimal air quality. Solazzo et al. (Solazzo et al., 2011) found both the highest





observed and modeled NOx concentrations inside a street canyon under a "quasi-parallel" situation.
Mumovic et al.(Mumovic et al., 2006) also suggested an accumulation effects along those canyons whose
axes are parallel to the wind direction.

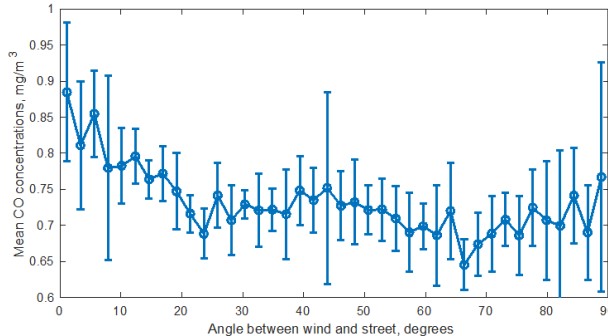

Figure 9. Influence of the angle between the directions of the wind and the street on CO concentrations.
Wind speed is specified as 6.5 m/s with emissions as that during rush hours.

### 3.3.3 Building heights

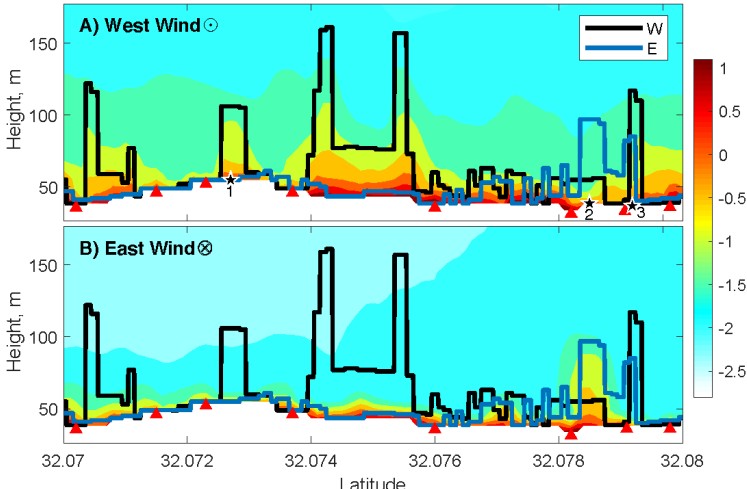

Figure 10. Spatial distribution of CO concentrations under west (A) and east (B) wind directions (3 m/s) in
latitude-height cross sections along Zhongyang Road during rush hours (marked as red line 2 in Figure 3B).
The outlines of buildings on both sides of the road are shown as black (west side) and blue (east side) lines.
Red triangles show the locations of major road intersections. Concentrations are shown in a natural
logarithm scale.

The influence of street and wind directions on modeled CO concentrations is more obvious in a latitude-
height cross section along a north-south direction street (Figure 10). Figure 11 shows the CO concentrations
in three longitude-height cross sections (marked as 1, 2, and 3 in Figure 10A) to illustrate the leakage
plumes from gaps between buildings. The modeled CO concentrations decrease sharply with height, as the
sources are from near the ground(Fu et al., 2017). The buildings in the east side of this road that is close to
the lake are lower than those in the west. The modeled CO concentrations are extended to a higher altitude
behind the tall buildings under west wind conditions (Figure 10A). The upwind buildings cause wake flows
that transport pollutant toward the buildings at pedestrian level and make an accumulation zone at the
leeward corners (Figure 11A and 11D). By contrast, the traffic-related emissions are not elevated to a
higher altitude with east wind due to the short buildings on that side (Figure 10B). Buildings located at

downwind of emission sources tend to create a flow pattern that blows pollutant away from them near the ground (Figure 11B and 11C). Previous studies also found similar concentration gradients between leeward

and windward of buildings when wind direction is perpendicular to the street canyon(Fu et al., 2017; Mumovic et al., 2006; Solazzo et al., 2011). For example, Fu et al.(Fu et al., 2017) found that pollutants emitted inside the street canyon with lower building heights in leeward than windward tend to disperse out of the canyon, and vice versa. When buildings exist both sides of the street, the flow and concentration distributions are largely determined by which side the taller building locates (Figure 11E and 11F). The

concentrations inside the street canyon are higher if the upwind building is taller than the downwind one.

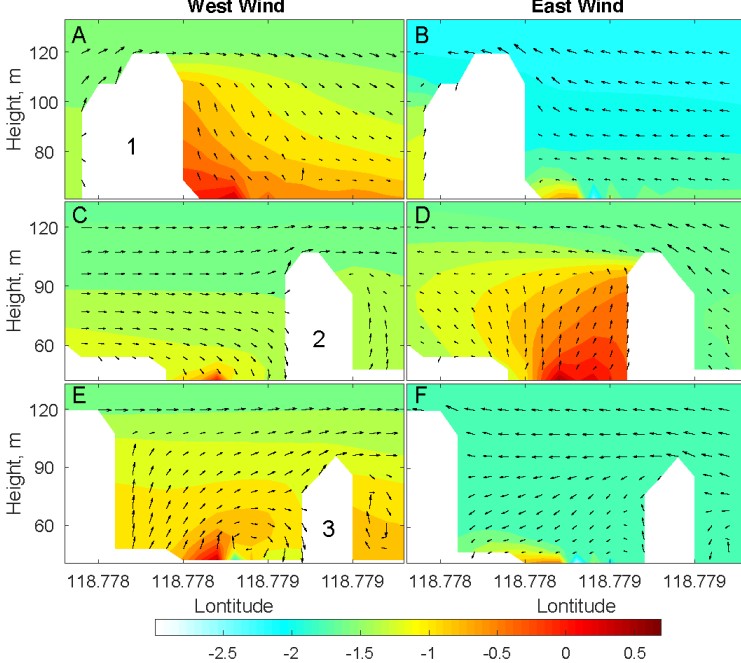

Figure 11. Spatial distribution of CO concentrations and wind vectors in longitude-height cross sections along three buildings in Zhongyang Road (marked as stars in Figure 10A). The concentrations distributions under west (A, C, and E) and east (B, D, and F) winds are shown. Note the color bar is in a natural

logarithm scale, and the vertical velocity is scaled by a factor of 2.5.

We also evaluate the relationship between the mean CO concentrations and the building heights in the upwind and downwind side of the street canyon in the whole model domain (Figure 12). We find the existence of upwind buildings generally increases the CO concentrations inside the street canyon compared to cases without buildings in that direction (i.e. zero building height) (Figure 12A). As discussed above, this

is associated with the wake flow pattern of the building (Figure 11A, 11D and 11E). The concentrations show no significant difference when the upwind building are ~10-45 m height, but decrease when building height further increases (Figure 12A). The influence of downwind building heights are largely monotonically with lower concentrations for higher heights. The interaction of upwind and downwind building heights is evaluated by their differences (e.g. upwind - downwind heights). Overall, the

concentrations are higher over streets canyons with higher upwind buildings, but the enhancement in concentrations begin to decrease if the difference is larger than ~30 m, consistent with Figure 12A. Similarly, higher downwind buildings bring down the concentrations inside the canyon monotonically, consistent with Figure 12B.

Figure 12C illustrates the influence of the variation of building heights within 50 m distance on the CO

concentrations. It indicates that the concentrations first increase when the standard deviation of building heights increase from 0 to ~10 m, reflecting the trapping effect of upwind buildings compared to flat surfaces. The concentrations significantly decrease when the nearby buildings are more variable. The

variation in building heights has been demonstrated to increase the ventilation rates and the vertical turbulent flux density, which helps to lower pedestrian-level pollution(Kurppa et al., 2018). Fu et al.(Fu et al., 2017) also found the concentration inside the street canyon first increase with the symmetric index of building heights, but decreases when the index becomes larger. These results suggest putting higher building in the prevailing downwind side of a road with large variability in building heights and multiple gaps between them generate the best pedestrian-level air quality.

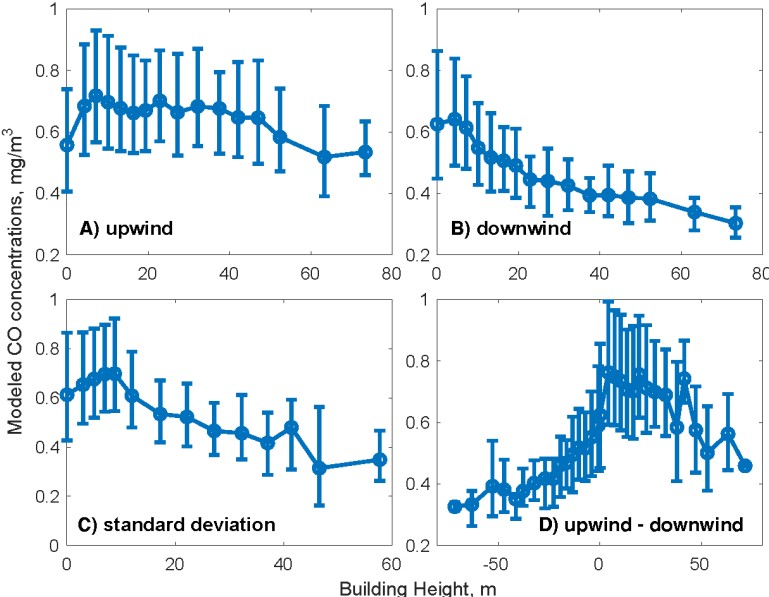

Figure 12. Relationship between geometric mean CO concentrations and building heights in the upwind (A), downwind (B) directions, (C) the standard deviations of nearby (within 50 m distance) building heights, and (D) the difference between the upwind and downwind building heights. Wind speed is assumed to be 6.5 m/s and emissions are specified as that during rush hours.

### 3.3.4 Distance to major roads

As discussed above, the modeled CO concentrations are higher inside the road network than outside of it. Figure 3 shows a clear decreasing trend of modeled concentrations from the road network to residential regions far away from the major roads. We thus calculate the distance from a given location to the nearest major roads ($d$), which include highways and arterial roads with emissions considered in this study (Figure 2B). Figure 13 shows the mean modeled CO concentrations ($C$) as a function of $d$. We used an exponential equation to fit this function: $C(d) = \alpha + \beta \exp(- d / k)$ following Apte et al.(Apte et al., 2017), where $\alpha$ represents the modeled background contribution from traffic-related sources, i.e. $C(\infty)$, $\beta$ is the sensitivity of C to $d$, and $k$ represents the spatial scale of the decay of $C$. We find the $\alpha$ value decreases as wind speed increases, indicating lower background values with higher wind speed as discussed in the section 3.3.1. Similarly, the $\beta$ values also decrease with higher winds. However, we find nearly identical $k$ values for all the wind speeds, suggesting that it is a universal parameter controlled by the atmospheric lifetime of pollutants but not influenced by meteorological conditions. Indeed, Apte et al.(Apte et al., 2017) also found different k values for NO, BC, and $NO_2$. Our $k$ values are much smaller than those calculated by Apte et al.(Apte et al., 2017) because they only consider the distance to the nearest highways and their $d$ values are much larger than ours. Our calculates are close to the model results of Biggart et al.(Biggart et al., 2019) that $NO_2$ concentrations also become quasi-stable ~50 m away from a major highway.





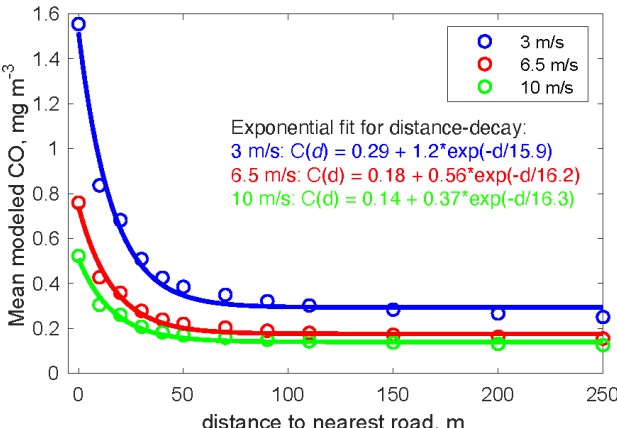

Figure 13. Relationship between the modeled CO concentrations and the distance to the nearest major roads (assuming east wind with emissions during rush hours).

## 4 Conclusions and Implications

This study demonstrates the potential of large eddy simulation in urban air quality modeling. Future directions of the model include a more dynamic emission inventory that considers real-time vehicle speed and traffic congestion(Pan et al., 2016). The model frame is also readily expandable to include other pollutant sources (e.g. point and area sources), multiple pollutants, and their chemical reactions(Wolf et al., 2020; Zhong et al., 2015). More realistic meteorological conditions possibly nudging from larger-scale weather and climate data could replace the limited number of assumed scenarios as adopted in this study(Heinze et al., 2016).

The revealed high-resolution spatial variability and its association with underlying meteorological conditions are useful for developing parameterization schemes for statistical models like AERMOD and ADMS-Urban, and land use regression models(Jerrett et al., 2005). As high-resolution information on urban building and traffic distribution is becoming more available, the approach could be relatively easily applied to other cities. The simulated tremendously high-resolution maps of concentrations in all major urban areas will be a vigorous part of smart city system(Silva et al., 2018), and serve as a data assimilation platform for many other products from satellite remote sensing and mobile platforms. The model results give hints for source contribution and hot-spot for urban air pollution, which could inform urban planning, air quality management, and risk mitigation. Combined with personal GPS data, the revealed very high-resolution of air quality map can inform epidemiological studies, health risk analysis, and alter personal behavior(Gao et al., 2019; Larkin and Hystad, 2017).

### Code/Data availability

The model code and validation data used in this work are available on the EBMG homepage: https://www.ebmg.online.

### Author contribution

YZ and NZ designed the research; YZ, XY, SW, and ZW performed model simulations; YZ, LD, SW, YM, MY, YL, and QL analyzed data; XH and HW provided emission data; SW, LW, XC, AD, LZ, and YX provided validation data; YZ, SW, LD, and HW wrote the paper.

### Competing interests

The authors declare that they have no conflict of interest.

### Acknowledgments

This study was supported by the National Key R&D Program of China (2019YFA0606803), Start-up fund of the Thousand Youth Talents Plan, Jiangsu Innovative and Entrepreneurial Talents Plan, and the Collaborative Innovation Center of Climate Change, Jiangsu Province. We are grateful to the High

Performance Computing Center (HPCC) of Nanjing University for doing the numerical calculations in this paper on its blade cluster system. We thank Rong Ye and Liang Luo for sample collection.

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
