# Peer review of "Large-eddy simulation of traffic-related air pollution at a very high-resolution in a mega-city: Evaluation against mobile sensors and insights for influencing factors"

_Atmospheric Chemistry and Physics, 2020_

## Referee Comment (RC1) · Anonymous Referee #1 · 22 Nov 2020

The authors have submitted a research article describing large eddy simulation of the atmospheric flow and CO concentration over the city of Nanjing. Numerical results are compared with experimental ones. The quality of the paper is good. A few minor improvements could be considered, as exposed below.

A - Specific comments :

1. Around line 104, the authors clarify the following : neutral stratification is assumed. It would be interesting to describe the meteorological conditions associated with the experiments : was the stratification mostly stable / unstable / neutral ?

- 2. Around line 106, the authors clarify the following : dry and wet deposition are neglected. Did the authors removed the experimental samples associated with rain periods ?
- 3. The temporal window (year / month / day / hour) associated with the experiment seems to be missing
- 4. The boundary conditions are poorly described. Boundary conditions used at the inlet / outlet / ground and at the top of the domain for velocity, pressure and CO should be clearly described. Regarding the inlet, is the flow steady ? If not, are the authors using synthetic turbulence ? Is the simulation considered fully resolved at the ground level ? If not, are the authors using some wall-functions ?
- 5. Regarding domain extension, is the domain high enough so that the boundary condition used at the top of the domain has no significant impact on the flow of air and on the CO concentration ?
- 6. The authors have not investigated the mesh resolution. It would be interesting to perform a simulation on a finer mesh to improve local estimations of the air flow and CO concentration. On the other hand, the comparison with experiments is made on a coarse grid. Thus, a simulation performed on a coarser mesh might perform as well as the present one compared with experiment. Another improvement could be obtained by simulating more wind directions.
- 7. Figure 13 is a bit misleading. The averaged concentration and the exponential curve fit very well. Each sample of the averaged concentration should be given with error bars (obtained from the simulation). On one hand, it is correct to claim that after averaging, the decay of concentrations associated with the distance

to the nearest major roads does not really depends on the wind speed, for the investigated cases. On the other, this does not rule out the possibility that, for a given wind direction and for a few streets, the decay will be much faster or much slower. The inclusion of error bars on the averaged concentration will allow the reader to visualize this variability.

- B Technical corrections :
  - 1. Line 58, the authors claim that trees increase turbulence and reduce concentrations. Trees are also associated with reduced street ventilation, which leads to higher concentrations. Thus, their effect remains controversial. https://doi.org/10.1016/j.envpol.2012.10.021
  - 2. Around line 140-155, the time dependence of the traffic flow and city-level emission should be clarified.
  - 3. The URL links given at the end of Section 2 are not in english, and are thus of limited use for an international audience
  - 4. In section 3.1 and at several other locations, the authors should replace "Air quality" by "Modeled CO concentration".
  - 5. Regarding Figure 3, it seems that the concentration is not exactly zero at the topright corner of the domain. This seems inconsistant with zero concentration at the inlet and no emission at this location.
  - 6. In Figure 3 but also across the paper, the authors often present CO concentrations without giving the associated elevation. Please clarify when ground-level concentrations are used.
  - 7. Figure 4 is of relatively poor quality and seems to contain negative modeled CO concentrations...

- 8. Figure 5 seems to contain a cluster of samples with measured CO concentrations around 0.2 and modeled concentrations between 1.2 and 1.8. On one hand, the cluster might not be statistically significant. On the other, it could be interesting to investigate it.
- 9. Figure 11 contains a typo on Longitude
- 10. Figure 12 should be updated. The abscissa is the building height for frames (A), (B) and (D) only.

---

## Referee Comment (RC2) · Anonymous Referee #2 · 17 Dec 2020

General Comments

This paper presents high-resolution modeling of CO concentrations in a high population urban area and a model performance evaluation based on high time-resolution observations. The research in this paper is a solid scientific study that adds to the knowledge we have of the variability in air concentrations in large urban areas. Below I detail some specific comments that should be addressed by the authors as well as some technical corrections.

[Figure]

Specific Comments

-Lines 183-185: please provide more reasoning for your decision to use the minimum CO concentration from the nine air quality monitoring stations. Are any of these stations located away from traffic/industrial sources or upwind of the city? Are any of these truly representative of a background concentration?

-Figure 4: I can't understand this plot at all. Why are all of the modeled CO concentrations negative? Why don't the peaks in the yellow and orange lines match the stated geometric means of 0.17 and 0.28 mg/m3? What do the blue and red portions of the F(C) equation represent? If the black lines represent the total frequency of residential streets + highways, why is the black curve it so similar to the yellow curve with no obvious influence from the orange curve?

-Figure 5 and 221-225: please provide detailed information on the time resolution of the modeled and monitored data used in this plot and the stated statistics. From the methods section, it appears that the model has a time resolution of 6 s, but the taxi data have a resolution of 10 s. How were the data transformed to be of equal time interval?

-Lines 223-232: the uncertainty for both the modeled and measured CO concentrations are a large percentage of the calculated 0.90 and 0.92 mg/m3 attributed to traffic sources. Combining this with my comment above that in-city monitoring sites may not be the best sources of background CO concentrations leads to the conclusion that the 40% attribution to traffic-related sources is very uncertain at best. I recommend adding further details on the uncertainty of this estimate.

Technical Corrections

-Lines 48-50: please add a citation for this sentence. While the point being made is generally true (i.e., there are few sensors in most major cities), the specific numbers quoted in this sentence must be attributed to the correct location. Also, consider

changing the beginning of this sentence to, "For example, in [city], . . ."

-Line 67: please clarify that CALPUFF is a puff model, not a Gaussian model.

-Line 76: the word "dynamics" should be added between "fluid" and "models".

-Figure 2: this figure is difficult to see and would be improved if it were higher resolution and/or a different color scheme.

-Figure 4: the legend is missing from this figure. Please include legend definitions for all three items plotted.

-Figure 6: panel Q needs to be clarified. Why is there a legend (mg/m3) on the right-hand side? Also, the explanation of the blue, red, and yellow bars does not make sense. R2 values compare the model and station, so there cannot be separate R2 values for the model and the station (i.e., the red and yellow bars).

-Figure 8 and associated text: are these ground-level concentrations or concentrations at 1.5 m (which would match the taxi data)?

-Figure 10. This plot would be improved by using actual concentrations rather than the natural log of concentrations. Using the natural log is not intuitive, as values <1 mg/m3 are negative.

-Figure 11: "longitude" is misspelled. As with Figure 10, concentrations would be a more intuitive item to plot, compared to the natural log of concentrations.
* * *

---

## Author Comment (AC1) · 28 Dec 2020

The authors have submitted a research article describing large eddy simulation of the atmospheric flow and CO concentration over the city of Nanjing. Numerical results are compared with experimental ones. The quality of the paper is good. A few minor improvements could be considered, as exposed below.

We thank the reviewer for the acknowledgment and the helpful comments and sugges-

[Figure]

tions. Please find our responses below.

A - Specific comments: 1. Around line 104, the authors clarify the following: neutral stratification is assumed. It would be interesting to describe the meteorological conditions associated with the experiments: was the stratification mostly stable / unstable / neutral?

We added the following sentence to elaborate it: "The actual vertical stability varies at Nanjing driven by the nocturnal cycle and large-scale weather patterns but with the neutral condition being the most frequent (Li, 2010). A neutral stratification is also considered as the most representative condition because stable and unstable conditions are either unfavorable or favorable for pollutant dispersion (Kurppa et al., 2018)."

2. Around line 106, the authors clarify the following: dry and wet deposition are neglected. Did the authors removed the experimental samples associated with rain periods?

No. Because CO is not dissolvable, and we don't find significant differences between CO levels on rainy and non-rainy days. We clarified this by adding this sentence in line 210: "We include data on both rainy and non-rainy days as CO is not dissolvable."

3. The temporal window (year / month / day / hour) associated with the experiment seems to be missing.

To clarify this, the sentence in line 131-134 was modified as: "Due to the large computational cost associated with model simulation, we don't run the model for a consecutive time window with actual meteorological conditions. Instead, we choose a selective combination of meteorological scenarios to represent the variability of meteorological conditions at Nanjing."

Also, we added the time window in line 181-182 for the evaluation dataset: "We evaluate the model results with observations collected from a mobile platform that is performed during September 2019 - October 2020."

4. The boundary conditions are poorly described. Boundary conditions used at the inlet / outlet / ground and at the top of the domain for velocity, pressure and CO should be clearly described. Regarding the inlet, is the flow steady? If not, are the authors using synthetic turbulence? Is the simulation considered fully resolved at the ground level? If not, are the authors using some wall-functions?

We clarified these settings by modifying the sentence in line 115 as: "A "Neumann" type boundary condition is applied for CO at the top and bottom of the model domain. A "Cyclic" type is used for its lateral boundary conditions, which yields an infinite and periodically repeating model domain. This is a reasonable assumption as our model domain only covers a portion of the city of Nanjing. For the horizontal wind and pressure, we use a "Dirichlet" type top boundary condition, a "No-slip" condition for the bottom and solid walls, and a "Cyclic" condition for the lateral boundary of the model domain. The flow is assumed to be steady at the inlet. The model explicitly resolves solid obstacles (e.g., buildings) on the Cartesian grid and reduces the 3D obstacle dimension to a 2D topography conforming to the Digital Elevation Model (DEM) format (Letzel et al., 2008)."

5. Regarding domain extension, is the domain high enough so that the boundary condition used at the top of the domain has no significant impact on the flow of air and on the CO concentration?

We clarified this by adding the following sentence in line 133: "Further increasing the model domain height (e.g., to 2000 m) has no significant impact on the modeled airflow and CO concentrations near the ground as most of the buildings are lower than 150 m."

6. The authors have not investigated the mesh resolution. It would be interesting to perform a simulation on a finer mesh to improve local estimations of the air flow and CO concentration.

The current resolution is the highest one we can achieve, which is limited by the reso-

lution of the building data. We clarified this by modifying the sentence in line 140 as: "The building data includes the geographical location of the outer shape of buildings ($0.0001° \times 0.0001°$ resolution) . . ."

On the other hand, the comparison with experiments is made on a coarse grid. Thus, a simulation performed on a coarser mesh might perform as well as the present one compared with experiment.

We thank the reviewer for bringing this up. That is indeed a drawback of our study. The relatively low sampling frequency of the sampler (10 s, equivalent to $\sim$100 m) limits us to compare them at a higher resolution. We acknowledged this point by adding this sentence in line 250: "We aggregate the model results into a 100 m resolution due to the relatively low sampling frequency of the mobile sampler (10 s, equivalent to $\sim$100 m), which is indeed a drawback and can be improved by higher time-frequency sensors."

Another improvement could be obtained by simulating more wind directions.

We agree with the reviewer. We have 48 scenarios (8 wind directions, 3 wind speeds, and 2 emission scenarios). That is what our budget allows us to do the most, as each scenario is computationally expensive. We, therefore, treat our simulation as demonstrative and could be improved by setting up more scenarios. We acknowledge this by adding the following sentence in line 151: "This results in a total of 48 scenarios, which is limited by our computational capacity. We thus consider our study as a demonstration of the model approach and it can be improved by more scenarios."

7. Figure 13 is a bit misleading. The averaged concentration and the exponential curve fit very well. Each sample of the averaged concentration should be given with error bars (obtained from the simulation). On one hand, it is correct to claim that after averaging, the decay of concentrations associated with the distance to the nearest major roads does not really depends on the wind speed, for the investigated cases. On the other, this does not rule out the possibility that, for a given wind direction and for a

few streets, the decay will be much faster or much slower. The inclusion of error bars on the averaged concentration will allow the reader to visualize this variability.

We added error bars in Figure 13 as suggested. The following sentence was added to the legend of this figure: "The circles and error bars are means and standard deviations, respectively."

The sentence in line 413 was modified as: "... modeled ground-level CO concentrations (C) and their standard deviations as a function ..."

A sentence was added to line 417: "The equation fits the modeled means well, despite the relatively large standard deviations especially when d is less than ∼50 m (Figure 13)."

B - Technical corrections: 1. Line 58, the authors claim that trees increase turbulence and reduce concentrations. Trees are also associated with reduced street ventilation, which leads to higher concentrations. Thus, their effect remains controversial. https://doi.org/10.1016/j.envpol.2012.10.021

"While trees are also associated with reduced street ventilation, which leads to higher pollutant concentrations (Vos et al., 2013)" was added in line 61.

2. Around line 140-155, the time dependence of the traffic flow and city-level emission should be clarified.

We added the following sentences at the end of this paragraph: "The diurnal variation of the traffic flows and subsequently the traffic emissions at Nanjing are based on the Gaode Map (https://report.amap.com/detail.do?city=320100). The total traffic CO emissions in the model domain are 0.77 and 0.60 kg s-1 for rush and non-rush hours, respectively."

3. The URL links given at the end of Section 2 are not in English, and are thus of limited use for an international audience.

We replaced the URL with its English version: "http://data.cma.cn/en".

4. In section 3.1 and at several other locations, the authors should replace "Air quality" by "Modeled CO concentration".

We modified the title of section 3.1 as "Very High-Resolution Modeled CO Concentration" We also modified the sentence in line 55 as "Here we present a very high-resolution air quality model for traffic-related CO air pollution in urban regions using large-eddy simulation" Similarly, the sentences in line 93: "a very high spatial resolution (less than 10 m) model for traffic-related CO air quality"; and in line 129: "To represent the CO air quality at pedestrian-level".

5. Regarding Figure 3, it seems that the concentration is not exactly zero at the top right corner of the domain. This seems inconsistent with zero concentration at the inlet and no emission at this location.

Thanks for pointing it out. We clarify this by adding the following sentences in line 116: "A "Cyclic" type is used for its lateral boundary conditions, which yields an infinite and periodically repeating model domain. This is a reasonable assumption as our model domain only covers a portion of the city of Nanjing."

6. In Figure 3 but also across the paper, the authors often present CO concentrations without giving the associated elevation. Please clarify when ground-level concentrations are used.

We added "ground-level" to describe the CO concentrations throughout the paper.

7. Figure 4 is of relatively poor quality and seems to contain negative modeled CO concentrations...

The x-axis is on log-scale. We modified the tick labels to their actual values instead of their log values.

8. Figure 5 seems to contain a cluster of samples with measured CO concentrations

around 0.2 and modeled concentrations between 1.2 and 1.8. On one hand, the cluster might not be statistically significant. On the other, it could be interesting to investigate it.

We thank the reviewer for bringing it up. We added an interesting discussion for this point in line 256: "The model tends to overestimate the measured CO concentrations over the Neihuanxi Line (the line of points on the left of Figure 5, location marked in Figure 1), which is a viaduct with better ventilation than ground-level roads. However, our model considers all the emissions at the ground-level thus simulates much higher concentrations than observations over this line. This also demonstrates the significant air quality benefit of building viaduct in an urban environment."

9. Figure 11 contains a typo on Longitude

We corrected this typo.

10. Figure 12 should be updated. The abscissa is the building height for frames (A), (B) and (D) only.

We modified the x-axis labels of the four panels.

**Fig. 1.**

[Figure]

**Fig. 2.**

A

B

C

0  1  2 km

0   0.1   0.2   0.3   0.4   0.5

**Fig. 3.**

Gaussian fit for frequency:

$$F(C) = 4.2 \times 10^3 \exp(-((C+0.55)/0.40)^2)$$
$$+ 2.0 \times 10^4 \exp(-((C+0.77)/0.18)^2)$$
$$R^2 = 0.99$$

□ Total Frequency
■ Mode 1: Residential roads
■ Mode 2: Other roads

**Fig. 4.**

[Figure]

**Fig. 5.**

**Fig. 6.**

A) 04/26/2020 Obs: 0.88

B) Mod+Sta:0.36+0.50

C) 04/27/2020 Obs: 0.58

D) Mod+Sta:0.32+0.31

**Fig. 7.**

[Figure]

**Fig. 8.**

[Figure]

**Fig. 9.**

[Figure]

**Fig. 10.**

West Wind       East Wind

**Fig. 11.**

Fig. 12.

Fig. 13.

- 3 m/s (blue)
- 6.5 m/s (red)
- 10 m/s (green)

Exponential fit for distance-decay:
3 m/s: $C(d) = 0.29 + 1.2 \cdot \exp(-d/15.9)$
6.5 m/s: $C(d) = 0.18 + 0.56 \cdot \exp(-d/16.2)$
10 m/s: $C(d) = 0.14 + 0.37 \cdot \exp(-d/16.3)$

---

## Author Comment (AC2) · 28 Dec 2020

Anonymous Referee #2 General Comments This paper presents high-resolution modeling of CO concentrations in a high population urban area and a model performance evaluation based on high time-resolution observations. The research in this paper is a solid scientific study that adds to the knowledge we have of the variability in air concentrations in large urban areas. Below I detail some specific comments that should be addressed by the authors as well as some technical corrections.

[Figure]

We thank the reviewer for the acknowledgment and the helpful comments and suggestions. Please find our responses below.

Specific Comments -Lines 183-185: please provide more reasoning for your decision to use the minimum CO concentration from the nine air quality monitoring stations. Are any of these stations located away from traffic/industrial sources or upwind of the city? Are any of these truly representative of a background concentration?

We clarified this by adding the following sentences in line 206: "Seven of these stations are located inside the model domain representing different functioning districts of the city. The remaining two are located at the suburbs to the west and northeast of the city center, which could be a reasonable representative for background concentrations depending on wind directions."

-Figure 4: I can't understand this plot at all. Why are all of the modeled CO concentrations negative? Why don't the peaks in the yellow and orange lines match the stated geometric means of 0.17 and 0.28 mg/m3? What do the blue and red portions of the F(C) equation represent? If the black lines represent the total frequency of residential streets + highways, why is the black curve it so similar to the yellow curve with no obvious influence from the orange curve?

We apologize for this confusion. The x-axis is on a log scale. We modified the x-tick values to their actual concentrations instead of their log values.

We used a two-mode Gaussian function to fit the data, i.e. the actual distribution is the sum of two Gaussian functions. The blue and red portions of the F(C) equation represent the yellow and orange modes of Gaussian functions, respectively (sorry we messed up the colors). The overall distribution is dominated by the yellow curve mode, i.e. most of the points are residential streets.

We modified the sentence in 235 as: ". . . follow a two-mode Gaussian distribution (i.e. a sum of two Gaussian functions, Figure 4) . . ." We modified the color of the text in Figure

4 to make the connection between the curves and the modes clearer. We also added a legend to Figure 4. The following sentence was added to the legend of Figure 4: "The yellow (residential streets) and orange (arterial roads, highways, and the nearby regions) curves represent the two Gaussian modes."

-Figure 5 and 221-225: please provide detailed information on the time resolution of the modeled and monitored data used in this plot and the stated statistics. From the methods section, it appears that the model has a time resolution of 6 s, but the taxi data have a resolution of 10 s. How were the data transformed to be of equal time interval?

We compare the time-averaged concentrations only. We had a sentence in line 136: "Hourly average data is achieved and we use the results of the last hour for analysis."

We added a sentence in line 144 to further clarify: "Due to the large computational cost associated with model simulation, we don't run the model for a consecutive time window with actual meteorological conditions. Instead, we choose a selective combination of meteorological scenarios to represent the variability of meteorological conditions at Nanjing."

We also modified the sentence in line 249 as: "We sample the hourly-mean model results with the same location, emission level (rush or non-rush hours), and wind speed/directions as the observations."

-Lines 223-232: the uncertainty for both the modeled and measured CO concentrations are a large percentage of the calculated 0.90 and 0.92 mg/m3 attributed to traffic sources. Combining this with my comment above that in-city monitoring sites may not be the best sources of background CO concentrations leads to the conclusion that the 40% attribution to traffic-related sources is very uncertain at best. I recommend adding further details on the uncertainty of this estimate.

We agree that big uncertainty is associated with our estimate. So we deleted this part

of the discussion as this is irrelevant to our main point of this paragraph.

Technical Corrections -Lines 48-50: please add a citation for this sentence. While the point being made is generally true (i.e., there are few sensors in most major cities), the specific numbers quoted in this sentence must be attributed to the correct location. Also, consider changing the beginning of this sentence to, "For example, in [city],:::"

We added the following sentence in line 50: "For example, there are 9 national air quality stations in Nanjing (http://hbj.nanjing.gov.cn/), and 8 air quality monitors in the City of New York (https://www.epa.gov/outdoor-air-quality-data/interactive-map-air-quality-monitors)."

-Line 67: please clarify that CALPUFF is a puff model, not a Gaussian model.

We modified this sentence as: "Gaussian plume and puff models have been widely used in such purpose for a long history, e.g. regulatory models such as AERMOD and CALPUFF"

-Line 76: the word "dynamics" should be added between "fluid" and "models".

Revised as suggested.

-Figure 2: this figure is difficult to see and would be improved if it were higher resolution and/or a different color scheme.

We replaced it with a higher-resolution version.

-Figure 4: the legend is missing from this figure. Please include legend definitions for all three items plotted.

Revised as suggested.

-Figure 6: panel Q needs to be clarified. Why is there a legend (mg/m3) on the right hand side? Also, the explanation of the blue, red, and yellow bars does not make sense. $R^2$ values compare the model and station, so there cannot be separate $R^2$

values for the model and the station (i.e., the red and yellow bars).

We added the following sentence to the legend of Figure 6: "Note the color bar for panel A-P is in panel Q." The sentence in line 271-273 was modified as: "Blue bars represent the regression between measured and model + regional background, while red and yellow bars are for the measured vs model only and measured vs station data only, respectively."

-Figure 8 and associated text: are these ground-level concentrations or concentrations at 1.5 m (which would match the taxi data)?

It shows the modeled ground-level concentrations, i.e. the first layer (2 m thick, or 1 m high if you consider the middle point of the layer). We added "ground-level (0-2 m above ground)" or "ground-level" throughout the text to make it clear.

-Figure 10. This plot would be improved by using actual concentrations rather than the natural log of concentrations. Using the natural log is not intuitive, as values <1 mg/m3 are negative.

We tried but the concentration decreases rapidly with height, which makes it hard to identify different values. We, therefore, chose a log scale to highlight the vertical structure of CO concentrations. We also modified the color scale label text to the actual concentrations to avoid negative values.

-Figure 11: "longitude" is misspelled.

This typo was revised.

As with Figure 10, concentrations would be a more intuitive item to plot, compared to the natural log of concentrations.

We modified the color scale label text to the actual concentrations similar to Figure 10.

Fig. 1.

[Figure]

**Fig. 2.**

[Figure]

**Fig. 3.**

Gaussian fit for frequency:
$$F(C) = 4.2 \times 10^3 \exp(-((C+0.55)/0.40)^2) + 2.0 \times 10^4 \exp(-((C+0.77)/0.18)^2)$$
$$R^2 = 0.99$$

□ Total Frequency
■ Mode 1: Residential roads
■ Mode 2: Other roads

Y-axis: Frequency, $\times 10^4$

X-axis: Modeled CO concentrations, mg/m$^3$

**Fig. 4.**

Fig. 5.

**Fig. 6.**

A) 04/26/2020 Obs: 0.88

B) Mod+Sta:0.36+0.50

C) 04/27/2020 Obs: 0.58

D) Mod+Sta:0.32+0.31

**Fig. 7.**

[Figure]

**Fig. 8.**

[Figure]

**Fig. 9.**

**Fig. 10.**

[Figure]

West Wind      East Wind

**Fig. 11.**

[Figure]

Fig. 12.

[Figure]

Fig. 13.